

# Temperate southern Australian coastal waters are characterised by surprisingly high rates of nitrogen fixation and diversity of diazotrophs

Lauren F. Messer[1,2], Mark V. Brown[3], Paul D. Van Ruth[4], Mark Doubell[4] and Justin R. Seymour[1]

[1] Climate Change Cluster, University of Technology Sydney, Sydney, New South Wales, Australia
[2] Centre for Microbiome Research, School of Biomedical Sciences, Queensland University of Technology, Brisbane, QLD, Australia
[3] School of Environmental and Life Sciences, University of Newcastle, Callaghan, New South Wales, Australia
[4] Aquatic Sciences, South Australian Research and Development Institute, Adelaide, South Australia, Australia

Corresponding author
Justin R. Seymour,
Justin.Seymour@uts.edu.au

## ABSTRACT

Biological dinitrogen ($N_2$) fixation is one mechanism by which specific microorganisms (diazotrophs) can ameliorate nitrogen (N) limitation. Historically, rates of $N_2$ fixation were believed to be limited outside of the low nutrient tropical and subtropical open ocean; however, emerging evidence suggests that $N_2$ fixation is also a significant process within temperate coastal waters. Using a combination of amplicon sequencing, targeting the nitrogenase reductase gene (*nifH*), quantitative *nifH* PCR, and $^{15}N_2$ stable isotope tracer experiments, we investigated spatial patterns of diazotroph assemblage structure and $N_2$ fixation rates within the temperate coastal waters of southern Australia during Austral autumn and summer. Relative to previous studies in open ocean environments, including tropical northern Australia, and tropical and temperate estuaries, our results indicate that high rates of $N_2$ fixation (10–64 nmol $L^{-1}$ $d^{-1}$) can occur within the large inverse estuary Spencer Gulf, while comparatively low rates of $N_2$ fixation (2 nmol $L^{-1}$ $d^{-1}$) were observed in the adjacent continental shelf waters. Across the dataset, low concentrations of $NO_3/NO_2$ were significantly correlated with the highest $N_2$ fixation rates, suggesting that $N_2$ fixation could be an important source of new N in the region as dissolved inorganic N concentrations are typically limiting. Overall, the underlying diazotrophic community was dominated by *nifH* sequences from Cluster 1 unicellular cyanobacteria of the UCYN-A clade, as well as non-cyanobacterial diazotrophs related to *Pseudomonas stutzeri*, and Cluster 3 sulfate-reducing deltaproteobacteria. Diazotroph community composition was significantly influenced by salinity and $SiO_4$ concentrations, reflecting the transition from UCYN-A-dominated assemblages in the continental shelf waters, to Cluster 3-dominated assemblages in the hypersaline waters of the inverse estuary. Diverse, transitional diazotrophic communities, comprised of a mixture of UCYN-A and putative heterotrophic bacteria, were observed at the mouth and southern edge of Spencer Gulf, where the highest $N_2$ fixation rates were observed. In contrast to observations in other environments, no seasonal patterns in $N_2$ fixation rates and diazotroph community structure were apparent. Collectively, our findings are consistent with the emerging view that $N_2$

fixation within temperate coastal waters is a previously overlooked dynamic and potentially important component of the marine N cycle.

## INTRODUCTION

By providing a source of new nitrogen (N), dinitrogen ($N_2$) fixation, the microbially mediated conversion of $N_2$ gas to ammonia, represents a fundamental process within oligotrophic marine environments (*Eugster & Gruber, 2012*; *Karl et al., 2012*). Based on global ocean estimates, the activity of $N_2$ fixing microorganisms (termed diazotrophs) contributes approximately 160 Tg of new N to the ocean annually (*Wang et al., 2019*). However, the majority of empirical observations contributing towards global $N_2$ fixation estimates have been derived from tropical and subtropical oceanic gyres (*Luo et al., 2012*), which have traditionally been deemed the principal ecological niche for marine $N_2$ fixation due to the activity of large filamentous and heterocystous cyanobacterial diazotrophs (*Breitbarth, Oschlies & LaRoche, 2007*; *Goebel et al., 2010*; *Karl et al., 2002*). In contrast, temperate coastal habitats have generally been thought to be enriched in dissolved inorganic N derived from terrestrial and upwelled sources (*Jickells, 1998*), thereby restricting the niche for biological $N_2$ fixation.

Temperate coastal waters are some of the most productive regions on Earth, which have historically been believed to be fuelled by the influx of bioavailable nutrients from rivers, groundwater, and through the mixing of offshore waters (*Jickells, 1998*). Often these hydrodynamic properties result in the development and maintenance of relatively eutrophic conditions, which in combination with typically cool sea surface temperatures, resulted in the supposition that diazotrophic growth and activity, particularly for the large filamentous cyanobacterium *Trichodesmium sp.*, would be limited (*Breitbarth, Oschlies & LaRoche, 2007*; *Howarth et al., 1988*; *Knapp, 2012*). However, largely due to the newly recognised abundance and activity of non-cyanobacterial diazotrophs and unicellular cyanobacteria (UCYN) outside of the traditional oceanic niches of $N_2$ fixation, there has been a recent paradigm shift in the potential importance of $N_2$ fixation in temperate coastal regions, where annual $N_2$ fixation rates have been estimated to exceed 16 Tg N (*Tang et al., 2019*). Therefore, an enhanced understanding of $N_2$ fixation rates and patterns in diazotroph diversity and activity within temperate coastal habitats is required to inform models of marine N availability at regional and global scales.

The widespread application of molecular surveys targeting the gene encoding a subunit of the nitrogenase enzyme complex (*nifH*), have greatly expanded the known diversity and global distribution of numerically important diazotrophs (*Cornejo-Castillo et al., 2018*; *Farnelid et al., 2011*; *Langlois et al., 2015*; *Moisander et al., 2010*; *Zehr et al., 1998*; *Zehr, Carpenter & Villareal, 2000*; *Zehr et al., 2003*). For example, *nifH* containing UCYN clades, *Candidatus* Atelocyanobacterium thalassa (UCYN-A), UCYN-B, and UCYN-C,

and putative heterotrophic, non-cyanobacterial diazotrophs, from the gamma-, delta-, and alphaproteobacteria, have recently been detected throughout the major ocean basins (*Díez et al., 2012*; *Farnelid et al., 2013*; *Fernández-Méndez et al., 2016*; *Gradoville et al., 2017*; *Langlois et al., 2015*). Investigations into the ecology of these novel diazotrophs have revealed a range of physiologies and patterns of biogeography. Both free-living (e.g., UCYN-B, and C; *Zehr & Turner, 2001*; *Taniuchi et al., 2012*; *Stenegren et al., 2018*) and symbiotic (e.g., UCYN-A) UCYN groups have been identified, and evidence suggests a diversity of closely related sub-lineages are representative of distinct ecological niches typically associated with "open ocean" (e.g., UCYN-A1) and "coastal" (e.g., UCYN-A2) environments (*Cornejo-Castillo et al., 2018*; *Farnelid et al., 2016*; *Thompson et al., 2014*; *Farnelid et al., 2016*).

The isolation of non-cyanobacterial diazotrophs from specific habitats, such as the Peruvian oxygen minimum zone (*Martínez-Perez et al., 2018*), and estuarine waters of the Baltic Sea (*Bentzon-Tilia et al., 2014*; *Farnelid et al., 2014*), imply relatively specialised niches for these organisms. However, genomic analysis of isolates, and metagenome-assembled genomes from the alphaproteobacteria and Planctomycetes, have revealed the metabolic flexibility of these groups, particularly in regard to their N cycling capabilities (*Bentzon-Tilia, Severin & Hansen, 2015*; *Delmont et al., 2018*; *Martínez-Perez et al., 2018*). Non-cyanobacterial diazotrophs are distributed throughout tropical and temperate latitudes and are sometimes the dominant members of diazotrophic communities (*Bombar, Paerl & Riemann, 2016*; *Delmont et al., 2018*; *Langlois et al., 2015*; *Moisander et al., 2014*). Notably, both non-cyanobacterial diazotrophs and UCYN have recently been identified as important constituents of temperate and coastal diazotroph communities (*Bentzon-Tilia et al., 2015*; *Messer et al., 2015*; *Mulholland et al., 2012*; *Needoba et al., 2007*; *Rees, Gilbert & Kelly-Gerreyn, 2009*; *Shiozaki et al., 2015a*; *Short & Zehr, 2007*), with their presence often associated with high rates of $N_2$ fixation (*Bentzon-Tilia et al., 2015*; *Tang et al., 2019*).

In the coastal waters surrounding the Australian continent, bioavailable sources of N are regularly depleted (*Condie & Dunn, 2006*). Significant rates of $N_2$ fixation have been observed throughout much of the tropical northern seas surrounding Australia (*Bonnet et al., 2015*; *Messer et al., 2016*; *Messer et al., 2017*; *Montoya et al., 2004*; *Raes et al., 2014*) and in the subtropical waters of the eastern Indian Ocean (*Raes et al., 2015*). High levels of diversity in *nifH* phylotypes have been detected throughout these regions, including important contributions by *Trichodesmium erythraeum*, UCYN-A, and the gammaproteobacterial group, Gamma A (*Moisander et al., 2014*; *Bonnet et al., 2015*; *Messer et al., 2017*). In contrast, our understanding of the importance of $N_2$ fixation within the temperate waters along the southern coastline of Australia, which are dominated by large inverse estuaries, is severely limited.

Inverse estuaries represent unique ecosystems within the coastal zone of arid climates (*Eyre, 1998*), where an excess of evaporation over precipitation results in the formation of strong positive salinity gradients from marine at the mouth to hypersaline at the head (*Nunes Vaz, Lennon & Bowers, 1990*; *Pritchard, 1952*). In contrast to classical estuaries, inverse estuaries receive little to no freshwater input (*Eyre, 1998*; *Smith & Veeh, 1989*), and can become seasonally isolated from the adjacent continental shelf-waters when density
fronts restrict oceanic inflow at the mouth (*Petrusevics et al., 2011*). Consequently, inverse estuaries can experience relatively oligotrophic conditions, giving rise to nutrient limitation in some systems (*Middleton et al., 2013*; *Smith & Veeh, 1989*).

Previously, we detected the presence of UCYN-A sub-lineages, UCYN-A1 and UCYN-A2, within the inverse estuary Spencer Gulf (*Messer et al., 2015*), an ecologically and economically important region of the south Australian marine environment (*Deloitte Access Economics, 2017*). Despite the fact that Spencer Gulf is typically oligotrophic, and primary production is reportedly N limited (*Middleton et al., 2013*), productive aquaculture industries and commercial fisheries are housed within the region, and the shallow waters act as foraging and nursery grounds for >30 species of threatened, protected, and iconic marine macro-organisms (*Robbins et al., 2017*). Seagrass-based $N_2$ fixation has historically been suspected to be an important source of new N in the shallow upper region (*Smith & Veeh, 1989*), however, how pelagic productivity is maintained within the low-nutrient waters of Spencer Gulf remains an open question. To test the hypothesis that $N_2$ fixation is a significant process within the temperate coastal waters of southern Australia, we investigated spatial and seasonal dynamics of $N_2$ fixation activity, and diazotroph diversity, in Spencer Gulf and the surrounding shelf waters.

## MATERIALS & METHODS

### Sample collection

Surface seawater samples were collected from Spencer Gulf, a large inverse estuary within the South Australian Gulf System ($\sim$22,000 km$^2$), and from adjacent continental shelf waters. Spencer Gulf is characterised by steep gradients in sea surface temperatures and salinity and demonstrates marked differences in physicochemical characteristics during autumn/winter and spring/summer (*Nunes & Lennon, 1986*; *Nunes Vaz, Lennon & Bowers, 1990*; *Petrusevics, 1993*). Therefore, samples were collected during two contrasting seasons, in the Austral autumn between 28th April and 8th May 2014, and in the Austral summer between 2nd and 9th December 2014. Although considered oligotrophic, Spencer Gulf hosts productive aquaculture industries, which have been implicated in localised nutrient enrichment (*Fernandes et al., 2007*; *Lauer et al., 2009*). To capture local environmental variability, sampling was performed along a latitudinal gradient within the estuary, from an offshore site situated near Kangaroo Island [35.84S, 136.45E] on the continental shelf, through to the hypersaline region in the north of Spencer Gulf. Four locations inside Spencer Gulf were selected, including, Spencer Gulf mouth [35.25S, 136.69E] and three locations along the edge of the basin, southern Gulf [34.377S, 136.11E], mid-Gulf [33.92S, 136.58E] and northern Gulf [33.04S, 137.59E] (Fig. 1).

Sampling at the mouth and shelf sites were conducted from on-board the *RV Ngerin* in conjunction with routine monitoring for the Southern Australian node of the Integrated Marine Observing System (IMOS). Samples were collected at the Kangaroo Island National Reference Station (NRSKAI; referred to as "shelf" hereafter), and SAM8SG mooring locations (referred to as "mouth" hereafter) (*Lynch et al., 2014*). A shore-based sampling protocol was adopted for the southern, mid, and northern Gulf sampling sites, whereby
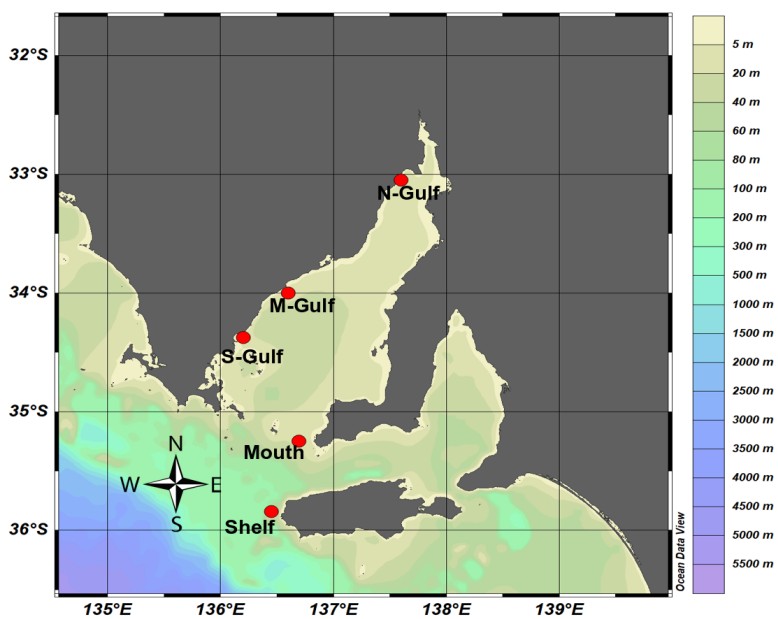

**Figure 1** **Sampling locations within Spencer Gulf and the adjacent continental shelf waters.** Samples were collected from the Kangaroo Island National Reference Station (Shelf), Spencer Gulf mouth (Mouth), south western edge (S-Gulf), mid western edge (M-Gulf), and northern Spencer Gulf (N-Gulf), with ocean bathymetry shown as a colour chart generated using Ocean Data View (*Schlitzer, 2018*).

surface samples were collected from jetties (piers), approximately 227, 154 and 440 m from the shore, respectively. In all cases, 60 L of water was collected from ~1 m below the surface using a plastic bucket. Buckets and sample storage carboys were rinsed three times with sample water prior to filling and washed with 10% HCl and MilliQ between stations. The temperature and salinity of each sample was immediately recorded using a multi-parameter portable meter (WTW Profiline Multi 3320; Xylem Analytics, Germany).

## Dissolved inorganic nutrient analyses

To determine ambient concentrations of dissolved inorganic nutrients, including $NO_3^-$ + $NO_2^-$, $PO_4^{3-}$ and $SiO_4^{4-}$ (hereafter referred to as $NO_3/NO_2$, $PO_4$, and $SiO_4$ respectively), subsamples (45 ml) were collected in triplicate 50 ml Falcon tubes from each sampling site. Samples were immediately frozen at −20 °C and kept frozen prior to analysis. A Flow Injection Analyser (Lachat QuikChem 8000) was used to determine concentrations of $NO_3/NO_2$, $PO_4$, $SiO_4$ in the <0.45 μm filtrate from each thawed sample, with a limit of detection of 0.01 μM.

## Particulate carbon, nitrogen, and $\delta^{15}$N analyses

To provide an estimate of the concentrations of particulate carbon (C) and N in the planktonic material and the natural abundance of the $^{15}$N isotope in particulate matter ($\delta$ $^{15}$N, $N_2$ fixation incubation $T_0$), subsamples of between 2–4 L of surface seawater were collected from each site. Subsamples were filtered onto GF/F grade 0.7 μm filters (Whatman, Kent, UK) which were previously individually packaged in aluminium foil and

pre-combusted at 450 °C for 4 h. Samples were stored double contained in two snap-lock bags and kept frozen at −20 °C, prior to being dried for 48 h at 60 °C. As previously described in *Messer et al. (2017)*, filters were analysed on an elemental analyser (Thermo Finnigan MAT Conflo IV) coupled to an isotope ratio mass spectrometer (IRMS; Thermo Finnigan Delta XP; limit of detection = 15 µg N per filter) at the Research Corporation of the University of Hawaii.

## Biological N$_2$ fixation incubations

To measure rates of N$_2$ fixation activity among planktonic diazotrophs, we performed stable isotope tracer addition experiments at each site with $^{15}$N-labelled N$_2$ gas. Acid-clean (10% HCl) 4 L Nalgene incubation bottles were rinsed three times with seawater from the site prior to being filled to over-flowing via silicone tubing, then capped with rubber septa head-space free. $^{15}$N$_2$ gas (3 ml, 98 atom%, Sigma-Aldrich, Australia, lot SZ1670V, 2013 batch) was injected into each incubation bottle prior to inversion 100 times to disperse the gas bubble.

Samples for whole community N$_2$ fixation (bulk seawater) and unicellular N$_2$ fixation (<10 µm size fraction) were incubated in triplicate at *in situ* sea surface temperature using aquaria heaters and water circulation pumps attached to an outdoor 60 L plastic incubator, which was exposed to a natural diurnal light cycle at surface seawater intensity. $^{15}$N$_2$ incubations were terminated via filtration after 24 h, by directly filtering the entire contents onto a pre-combusted (450 °C for 4 h; packaged in aluminium foil) GF/F grade 0.7 µm filter (Whatman; whole community), or through a 10 µm polycarbonate membrane filter (Isopore, EMD Merck Millipore, Billerica, MA, USA) onto a pre-combusted GF/F grade 0.7 µm filter (Whatman; unicellular size fraction). Enriched filters were stored double contained in two snap-lock bags to prevent any possible cross-contamination and kept frozen at −20 °C prior to analysis.

Following methods described in *Messer et al. (2017)*, $^{15}$N$_2$ amended GF/F filters were dried (60 °C for 48 h) separately to natural abundance samples to prevent cross-contamination. Total particulate N and C and isotopic composition were determined on an elemental analyser (Thermo Finnegan MAT Conflo IV) coupled to an IRMS (Thermo Finnegan Delta XP, limit of detection = 15 µg N per filter) at the Research Corporation of the University of Hawaii. Assimilation rates were calculated following *Montoya et al. (1996)*. An atom% enrichment equivalent to 75% of the theoretical for a 24 h incubation was used as the enrichment factor for volumetric rate calculations to account for the incomplete dissolution of the $^{15}$N$_2$ gas bubble (*Großkopf et al., 2012*; *Mohr et al., 2010*), following *Messer et al. (2017)*.

## Collection, preservation, and extraction of microbial nucleic acids

In order to concentrate microbial cells for nucleic acid extraction, amplicon sequencing, and quantitative PCR, triplicate 2 L samples were filtered onto 0.2 µm membrane filters (Durapore, EMD Merck Millipore). Filters were stored at −20 °C during the field sampling (∼2 weeks) and transported to the laboratory on dry ice before being stored at −80 °C until extraction. The MoBio PowerWater DNA isolation kit (MoBio Laboratories, Carlsbad,

CA, USA; now Qiagen) was used to extract microbial community DNA, following the manufacturer's guidelines including an additional incubation with solution PW1 (10 min at 60 °C) prior to 10 min of bead beating, to ensure complete cell lysis.

### *nifH* amplicon sequencing and analyses

To determine the diversity of diazotrophic bacterioplankton, a fragment of the *nifH* gene was amplified using a nested protocol and the degenerate primers nifH3 and nifH4, and nifH1 and nifH2 (*Zani et al., 2000*; *Zehr & Turner, 2001*), largely following methods previously described (*Messer et al., 2015*; *Messer et al., 2016*; *Messer et al., 2017*). The following PCR reaction conditions were used to amplify the *nifH* gene: 95 °C (2 min) followed by 30 cycles of 95 °C (1 min), 48 °C (1 min) and 72 °C (1 min) followed by 72 °C (10 min). Amplification was confirmed using gel electrophoresis, replicates were pooled, and the resultant fragment was sequenced using the 454 FLX Titanium pyrosequencing platform (Roche, Nutley, NJ, USA) at Molecular Research LP (Shallowater, TX, USA).

The Quantitative Insights Into Microbial Ecology (QIIME) (*Caporaso et al., 2010a*; *Caporaso et al., 2010b*) open source software was used to process *nifH* pyrosequencing reads. Briefly, sequences were de-multiplexed and the low-quality sequences were removed (q <25 and <200 bp in length) using default parameters. Chimeric sequences were removed using USEARCH61 with default parameters against an unaligned version of a curated *nifH* reference database exported from Arb (downloaded from: http://wwwzehr.pmc.ucsc.edu/nifH_Database_Public/; *Heller et al., 2014*; *Zehr et al., 2003*). The remaining high-quality reads were clustered at 99% sequence identity using UCLUST, whereby sequences within 1% identity of the most abundant read were classified as operational taxonomic units (OTUs; *Edgar, 2010*). A representative sequence set was generated based on the most abundant sequence comprising an OTU. The PyNAST aligner tool (*Caporaso et al., 2010a*) was used with default parameters to BLAST and pairwise align representative *nifH* OTU sequences to those from the aligned version of the same curated *nifH* database used for chimera removal, providing putative taxonomy and "best hits" to primarily uncultured environmental sequences (*Heller et al., 2014*; *Zehr et al., 2003*). Any potential stop codons and frameshifts in the *nifH* sequences were identified using the FrameBot tool from the FunGene pipeline using default parameters (*Fish et al., 2013*). As part of this pipeline, taxonomy was assigned to the closest representatives within the Ribosomal Database Project's *nifH* database based on amino acid identity (AAI) and sequence alignment (*Fish et al., 2013*). Finally, an OTU by sample matrix was generated, in which each sample was rarefied to the lowest number of sequences per sample (3,068) and singletons were removed prior to downstream analyses.

### Quantification of UCYN-A *nifH* genes

Based on our previous observations (*Messer et al., 2015*), we hypothesised that UCYN-A would be the most important diazotrophic group within Spencer Gulf and the adjacent continental shelf waters. In order to determine UCYN-A abundance, previously designed TaqMan qPCR probes (Table S1) were utilised to quantify the UCYN-A1 (*Langlois, Hummer & LaRoche, 2008*) and UCYN-A2 (*Thompson et al., 2014*) clades. qPCR standards

were either cloned into the P-Gem T Easy Vector (Promega, Sydney, NSW, Australia) following the manufacturer's guidelines (UCYN-A2) as previously described (*Messer et al., 2017*), or synthesised into the PUC-57 Amp (Genewiz) vector (UCYN-A1). The *nifH* gene inserts were then amplified from the plasmid DNA using plasmid specific PCR primers targeting the M13 binding site of the vector. A band of the correct size was purified from an electrophoresis gel using the Isolate II Gel/PCR Purification kit (Bioline, Eveleigh, NSW, Australia). DNA was then quantified using a Qubit Fluorometer and serially diluted to generate a standard curve incorporating $10^7$ to $10^1$ *nifH* copies.

qPCR reactions were performed as previously described in *Messer et al. (2017)*. Specifically, template DNA was diluted 1:5 using nucleic-acid-free $H_2O$ to prevent inhibition and 5 µl of the template dilution was subsequently used in the qPCR assay. Each qPCR reaction included 200 nM of forward and reverse primer, 100 nM of TaqMan probe, 2x TaqMan Master Mix II, and 3 µl of nucleic-acid-free $H_2O$. Samples were analysed in triplicate, with additional triplicate technical replicates and triplicate no template negative controls (5 µl nucleic-acid-free $H_2O$), alongside the relevant standards (also analysed in triplicate). Reaction conditions were optimised for each primer and probe set using a combination of temperature, annealing time, and extension time gradients on a StepOnePlus$^{TM}$ Real-Time PCR machine (software v2.3; Applied Biosystems, Thermo Fisher Scientific, Scoresby, Victoria, Australia). The final optimal reaction conditions were identified to be: 50 °C (5 min), 95 °C (10 min) and 40 cycles of 95 °C (15 s) and 64 °C (60 s) for UCYN-A2; and 95 °C (10 min) followed by 40 cycles of 95 °C (15 s), 55 °C (15 s) and 72 °C (60 s) for UCYN-A1. Linear regression analyses of quantification cycle (Cq) versus log10 *nifH* gene copies demonstrated that the UCYN-A2 assay had a mean $R^2$ of 0.999 and an efficiency between 99.2–99.9% and the UCYN-A1 assay had a mean $R^2$ of 0.993 and an efficiency between 92.0–98.7%. The Cq limit of detection and quantification for each assay was identified to be equivalent to ∼1–10 *nifH* copies per reaction.

## Statistical analyses

Prior to testing for significant differences between "season" and "site", environmental data, $N_2$ fixation rates, and qPCR data were checked for normality and homogeneity of variance using the Shapiro–Wilk and Brown-Forsythe tests respectively (SPSS, IMB Statistics 24). Data meeting these criteria were tested for significance using a one-way ANOVA, while a Kruskal Wallis ANOVA on ranks was used for data that failed to meet the stipulations of normality (SPSS, IMB Statistics 24). Pearson correlation coefficients and significance values were calculated (SPSS, IMB Statistics 24) between biological and environmental variable pairs across the entire dataset, and independently for samples collected in Austral autumn or summer.

Statistical analyses of diazotroph community dissimilarity were performed using the PRIMER 7 + PERMANOVA software. The final OTU by sample matrix was square-root transformed and a Bray Curtis resemblance matrix was generated. Significant differences between *nifH* amplicon sequencing profiles were explored using the non-parametric Analysis of Similarity (ANOSIM) test, using either "season" or "site" as a factor, while the contribution of each OTU to the observed dissimilarity between sampling sites was
determined using Similarity Percentage analysis (SIMPER). In addition, a distance-based linear model (DistLM) was generated from the Bray-Curtis resemblance matrix, using the corresponding site-specific environmental metadata as predictor variables. Relationships between the environmental predictor variables and diazotroph community composition were also investigated using the BEST, biota and environment (BIOENV) test, using Spearman rank correlation.

The multivariate relationships between individual diazotroph OTUs, environmental metadata, and $N_2$ fixation rates were explored using a negative binomial many-generalised linear model (*Wang et al., 2012*). The model was performed using the mvabund (v.4.1.3) package (*Wang et al., 2012*) in R (v4.0.2) and R studio (v1.3.959) (*R Core Team, 2013*). The *nifH* OTU by sample matrix was input as count data and converted to an mvabund object prior to model creation using the 'manyglm' function. The analysis of deviance table was generated using the 'anova' function with 'p.uni = adjusted' selected to correct for the effect of multiple testing.

## RESULTS

### Environmental characteristics of Spencer Gulf and shelf waters

Consistent with the inverse estuarine nature and seasonal variability of Spencer Gulf, patterns in sea surface temperature (SST) and salinity exhibited a clear transition from cooler oceanic conditions in southern shelf waters, to warmer and hypersaline conditions in the northern region of the Gulf (Fig. 1; Table 1). Across this gradient, SST ranged from 18 °C to ~23 °C, while salinity increased from 36 at the mouth to ≥ 40 at the northern site (Table 1). During the Austral autumn, SST was typically lower than SST observed during the summer (Table 1), with mean temperature (± standard deviation) across the five sites, 18.9 ± 0.8 °C relative to 21.0 ± 1.8 °C, respectively. In contrast, the salinity profile of Spencer Gulf was highly similar during both the Austral autumn and summer across the five sampling sites, with means for each season (± standard deviation) of 37.3 ± 1.65 and 37.1 ± 1.81, respectively.

Concentrations of dissolved inorganic nutrients were relatively stable between the southern shelf and northern Spencer Gulf waters. Indeed, $NO_3/NO_2$ concentrations were always <0.05 μM, and $PO_4$ concentrations were generally low, ranging from 0.01 (i.e., limit of detection) to 0.08 μM across the five sampling locations (Table 1). Mean (± standard deviation) $NO_3/NO_2$ and $PO_4$ concentrations were similar between the two sampling periods, at 0.02 ± 0.01 and 0.03 ± 0.02 μM during Austral autumn, and 0.03 ± 0.01 and 0.04 ± 0.03 μM during Austral summer, respectively. Conversely, concentrations of $SiO_4$ showed a sharp increase from the southern shelf to northern Gulf waters, ranging from 0.22 up to 1.10 μM (Table 1). While mean $SiO_4$ concentrations were typically elevated during Austral autumn compared to summer, at 0.49 ± 0.36 and 0.38 ± 0.14 μM, respectively.

### Biological $N_2$ fixation rates in temperate southern Australia

Measurable rates of $N_2$ fixation occurred at all sites during both the Austral autumn and summer, but rates were highly heterogeneous ranging from 2 nmol $L^{-1}$ $d^{-1}$ to 64 nmol $L^{-1}$ $d^{-1}$ (Fig. 2). Across the entire dataset, no significant differences were observed

**Table 1** **Physico-chemical metadata associated with each sampling site.**

| Sample | Sampling Time | Temp. (°C) | Salinity | NO$_3$/NO$_2$ (μM) | PO$_4$ (μM) | SiO$_4$ (μM) | PC (μg) | PN (μg) |
|---|---|---|---|---|---|---|---|---|
| A_Shelf | 14:30 | 18.9 | 36.0 | 0.04 | 0.03 | 0.22 | 373 | 50.3 |
| A_Mouth | 8:00 | 18.7 | 36.0 | 0.02 | 0.06 | 0.36 | 377.8 | 53.2 |
| A_S-Gulf | 15:00 | 18 | 37.0 | 0.01 | 0.01 | 0.25 | 419.6 | 54.6 |
| A_M-Gulf | 7:30 | 18.8 | 37.7 | 0.01 | 0.02 | 0.52 | 341.7 | 37.8 |
| A_N-Gulf | 16:00 | 20.1 | 40.0 | 0.04 | 0.03 | 1.1 | 389.9 | 47.9 |
| S_Shelf | 6:30 | 18.7 | 36.0 | 0.02 | 0.08 | 0.24 | 1184.2 | 36.7 |
| S_Mouth | 9:00 | 19.6 | 36.0 | 0.01 | 0.04 | 0.24 | 762.7 | 44.6 |
| S_S-Gulf | 15:50 | 22.3 | 36.5 | 0.03 | 0.02 | 0.52 | 376.8 | 45.5 |
| S_M-Gulf | 7:55 | 21.1 | 36.9 | 0.03 | 0.05 | 0.53 | 755.5 | 62.8 |
| S_N-Gulf | 16:00 | 23.1 | 40.3 | 0.04 | 0.02 | 0.39 | 342.3 | 42.8 |

**Notes.**

A, Autumn; S, Summer; Temp., sea surface temperature; PC, particulate carbon; PN, particulate nitrogen.
Sampling Time refers to the local time at the point of sample collection (Australian Eastern Standard Time).

between whole community (WC) and unicellular size fraction (USF) N$_2$ fixation rates (Kruskal-Wallis test, $H = 0.32$, d.f. $= 1$, $n = 30$, $P = 0.574$), indicating that the unicellular size fraction contributed the majority of the observed N$_2$ fixation activity. Overall, no significant differences in N$_2$ fixation rates were observed between incubations conducted during Austral autumn compared to summer (Kruskal-Wallis test, $H = 1.397$ and $1.931$, d.f. $= 1$, $P = 0.237$ and $0.165$, for WC and USF respectively; $n = 15$ per season). During both Austral autumn and summer, WC and USF N$_2$ fixation rates were highly correlated, with Pearson correlation coefficients (r) of 0.85 and 0.76 respectively, further supporting the proposition that the unicellular size fraction contributed the majority of the observed N$_2$ fixation activity.

When grouped by "site" as opposed to "season", N$_2$ fixation rates exhibited significant spatial heterogeneity (One-way ANOVA, $P \leq 0.001$, $F = 37.38$, d.f. $= 4$, $n = 6$ per site). The lowest rates of N$_2$ fixation during both the Austral autumn and summer occurred in the southern shelf waters, with maximum rates at this site reaching only $8 \pm 2$ nmol L$^{-1}$ d$^{-1}$ (mean $\pm$ standard deviation; Fig. 2). In contrast, N$_2$ fixation rates peaked in the waters at the mouth of Spencer Gulf, where they reached $64 \pm 3$ and $40 \pm 19$ nmol L$^{-1}$ d$^{-1}$, in Austral autumn and summer respectively (Fig. 2). Relative to rates observed at the mouth of Spencer Gulf, N$_2$ fixation rates decreased at the southern and mid-western sites of the gulf during both autumn and summer (Fig. 2). N$_2$ fixation rates then showed a notable increase at the northern-gulf site, reaching $29 \pm 4$ nmol L$^{-1}$ d$^{-1}$ during Austral autumn, and $14 \pm 10$ nmol L$^{-1}$ d$^{-1}$ during Austral summer (Fig. 2).

N$_2$ fixation rates were significantly correlated with low concentrations of NO$_3$/NO$_2$ (Pearson's r: -0.53; $P = 0.002$, $n = 30$, USF). This relationship was maintained when considering only Austral summer samples (r: -0.64; $P = 0.01$; $n = 15$, USF), but not when only considering those collected during Austral autumn. In contrast, during the Austral autumn N$_2$ fixation rates were positively correlated to PO$_4$ concentrations (r: 0.73; $P = 0.002$; $n = 15$, WC). No significant relationships were observed between N$_2$ fixation

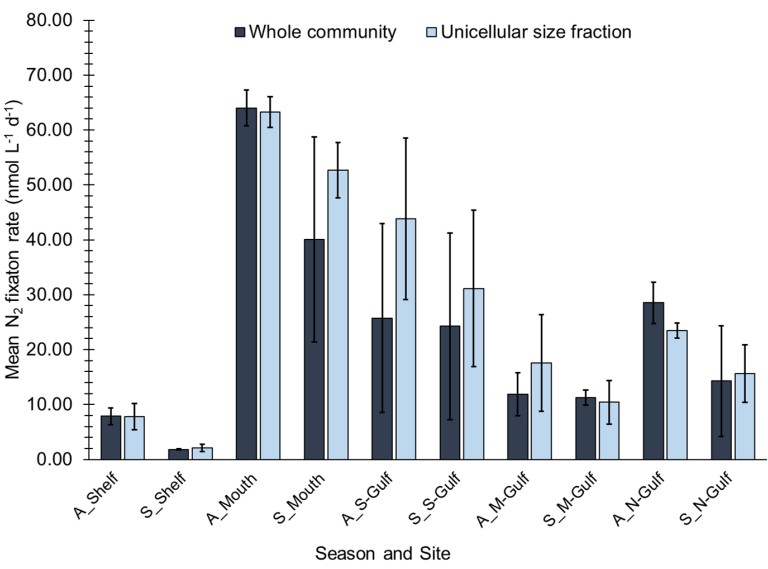

**Figure 2** **Biological N₂ fixation rates measured during Austral autumn and summer in south Australian coastal waters.** Rates have been corrected to account for the incomplete dissolution of the $^{15}N_2$ gas bubble (see Methods). Error bars represent the standard deviation about the mean ($n = 3$).

and SST or salinity, despite clear spatial gradients in these environmental parameters (Table 1).

## Diversity and composition of *nifH* containing bacterioplankton

After rarefaction to 3,068 sequences per sample and the removal of singletons, between 159 and 332 *nifH* OTUs were detected at each sampling site. The diversity of *nifH* containing bacterioplankton increased along the latitudinal gradient of Spencer Gulf, whereby Shannon's Diversity (H') was lowest in the southern shelf waters, where H' = 1.95 and 2.86 and peaked at the mid-western edge of Spencer Gulf, where H' = 4.97 and 4.37, during Austral autumn and summer respectively (Table S2). Despite the site-specific differences in diazotroph diversity, mean H' across the Gulf was approximately equal for both sampling seasons, whereby H' = 3.67 during austral autumn, and H' = 3.58 during austral summer.

Phylogenetic analyses of *nifH* sequences demonstrated that the most abundant OTUs ($n = 25$), equivalent to ~53% of total sequences and between 15 and 82% of sequences for any given sample, comprised a mixture of Cluster 1 and Cluster 3 diazotrophs at ≥ 83% amino acid identity (AAI; Table S3). A Bray-Curtis resemblance matrix of rarefied *nifH* sequence data was used to compare diazotroph community composition within and between the southern shelf waters and Spencer Gulf sampling locations, revealing significant spatial variability in diazotroph assemblage structure (ANOSIM, R: 0.59, $P = 0.005$). SIMPER analysis revealed 99.7% and 100% community dissimilarity between northern Gulf diazotroph assemblages and those in the shelf waters and at the mouth of the Gulf, respectively. Diazotroph assemblages in the shelf waters and mouth were dominated by five OTUs identified to be the UCYN-A1 open ocean ecotype (OTU51120, OTU3535,

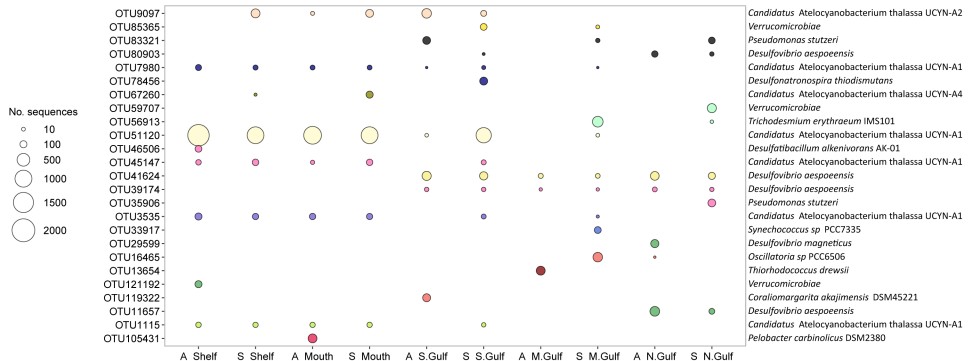

**Figure 3** Relative abundance of the top 25 *nifH* OTUs and their taxonomic assignment (closest representative) detected within south Australian coastal waters during Austral autumn (A_) and summer (S_).

OTU45147, OTU7980, and OTU1115; Fig. S1), which collectively represented 75% and 56% of sequences at the shelf during Austral autumn and summer respectively (Fig. 3). Similarly, these OTUs comprised 54% and 58% of sequences at the mouth during Austral autumn and summer (Fig. 3). Correspondingly, diazotroph communities in the southern shelf waters and at the mouth of the gulf shared the greatest similarity in composition, with SIMPER analysis revealing only 63.4% dissimilarity between these populations. The dissimilarity between the shelf waters and the mouth was largely driven by the coastal and open ocean ecotypes UCYN-A2 and UCYN-A4 (OTU9097 and OTU67260, respectively), which were collectively present at higher relative abundances at the mouth (Fig. 3).

Spencer Gulf communities showed a decline in the abundance of UCYN-A OTUs, and a greater proportion of sequences associated with non-cyanobacterial diazotrophs, along with a small proportion of OTUs closely related to filamentous cyanobacteria such as *Trichodesmium erythraeum* (Fig. 3; Table S3). The average relative abundance of two UCYN-A1 open ocean group OTUs, (OTU51120 and OTU3535), were identified by SIMPER analysis as the main drivers of community dissimilarity between the shelf waters, Spencer Gulf mouth, and northern gulf diazotroph assemblages. At the southern gulf site, a transitional community was observed, which comprised UCYN-A1 and UCYN-A2 (12–44% of sequences), *Pseudomonas stutzeri* (7%), *Desulfovibrio aespoeensis* (8–12%), *Coraliomargarita akajimensis* (7%), and *Desulfonatronospira thiodismutans* (7%). In contrast, at the northern site the community was primarily comprised of OTUs related to *Desulfovibrio aespoeensis* (10–28%), *Pseudomonas stutzeri* (11%), and Verrucomicrobiae (11%; Fig. 3). SIMPER analysis identified the *Desulfovibrio aespoeensis* OTU (OTU41624; 96% AAI similarity) as also being responsible for the between-site discrimination of the diazotroph community, with this OTU absent from assemblages detected in the southern waters. Interestingly, only a small proportion of the most abundant 25 OTUs were represented at the mid-Spencer Gulf site (15–38%) and northern-Spencer Gulf site (32–36%) especially during the Austral autumn. Instead, overall low abundance

OTUs, which were typically unique to these sites (i.e., OTUs representing <0.5% of total sequences), were responsible for the high alpha diversity associated with these sites.

Across the dataset, several variables were identified as having a significant effect on the relative abundance and composition of diazotrophic bacterioplankton within Spencer Gulf and the adjacent shelf waters. These included $NO_3/NO_2$ ($P = 0.002$), $N_2$ fixation by the unicellular size fraction ($P = 0.007$) and the whole community ($P = 0.016$), salinity ($P = 0.017$), temperature ($P = 0.037$), particulate nitrogen (PN; $P = 0.039$), and $PO_4$ ($P = 0.048$; Many GLM, Table S4). Only three of these predictors displayed significant relationships (adjusted $P$-value <0.1) with individual OTUs, including PN (1 OTU), $N_2$ fixation by the unicellular size fraction (14 OTUs), and salinity (4 OTUs; Table S4).

Approximately 33% of the spatial variation in diazotroph community dissimilarity could be explained by ambient salinity and $SiO_4$ concentrations (DistLM $R^2$: 0.33; salinity $F = 2.24$, $P = 0.001$; $SiO_4$ $F = 1.56$, $P = 0.028$; $n = 10$). The importance of salinity and $SiO_4$ in structuring the diazotroph community was further confirmed by BEST/BIOENV analyses, resulting in a significant ($P = 0.01$, $n = 10$) coefficient, Rho = 0.67, using Spearman's Rank correlation. The sequential addition of the environmental parameters, PN, $NO_3/NO_2$, and $PO_4$, reduced the strength of the correlation to 0.56, 0.52, and 0.49, respectively. In contrast to the observed spatial heterogeneity in diazotroph assemblage structure, no significant differences in diazotroph community dissimilarity were observed between the Austral autumn and summer sampling times (ANOSIM, R: $-0.12$, $P = 0.80$).

## Abundance of UCYN-A1 and UCYN-A2 *nifH* genes

qPCR derived abundances of UCYN-A1 and UCYN-A2 *nifH* genes demonstrated higher abundances of these organisms in shelf waters and at the more southern sites of Spencer Gulf (Fig. 4). Specifically, the maximum mean abundance of UCYN-A1 occurred in the southern shelf waters during Austral summer, whereby $5.4 \pm 4.7 \times 10^4$ *nifH* copies $L^{-1}$ were detected (Fig. 4). Similarly, UCYN-A2 also reached maximum abundance in the shelf waters during Austral summer, with mean *nifH* copies $1.9 \pm 1.4 \times 10^4$ $L^{-1}$ (Fig. 4). Across all sampling locations, UCYN-A1 was significantly more abundant during the Austral summer compared to autumn (Mann Whitney test, U: 69, $P < 0.05$). While UCYN-A2 abundances did not differ significantly between the Austral autumn and summer sampling.

Across the entire dataset, UCYN-A1 abundance was positively correlated with concentrations of $PO_4$ (r: 0.39; $P = 0.03$, $n = 30$). In contrast, overall UCYN-A2 abundance was not significantly correlated with any of the measured environmental parameters. However, when analysed by "season", UCYN-A2 abundance was negatively correlated to SST during both Austral autumn and summer ($n = 15$ per season; r: $-0.55$ and r: $-0.53$, $P = 0.03$ and 0.04, respectively). In addition, during Austral autumn UCYN-A2 abundance was negatively correlated to $PO_4$ concentrations (r: $-0.53$; $P = 0.04$; $n = 15$). Despite the potential importance of salinity in structuring the overall diazotroph community, no significant relationships were observed between UCYN-A qPCR derived abundances and salinity.

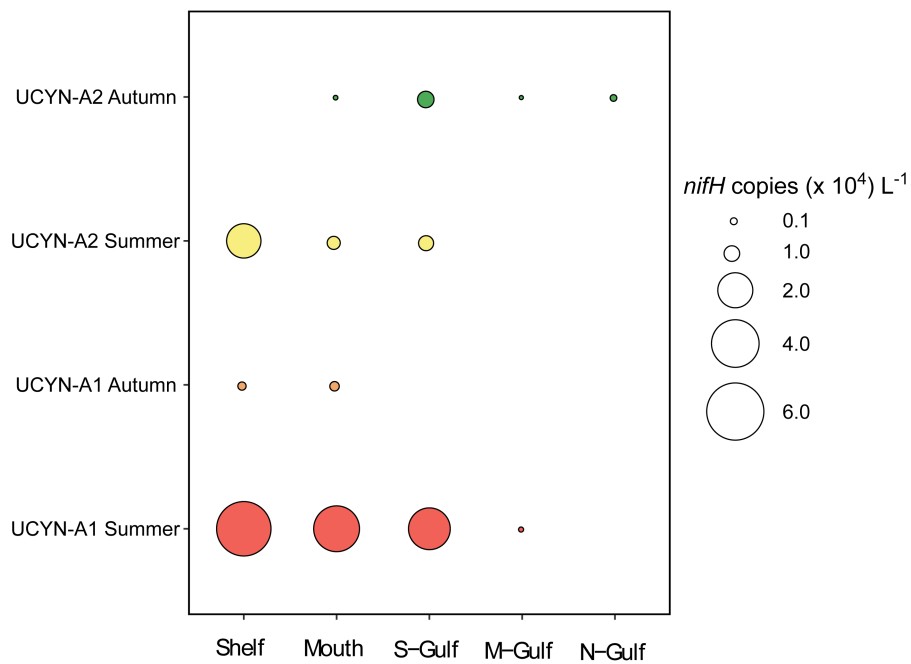

**Figure 4** Mean qPCR derived abundances of UCYN-A1 and UCYN-A2 in south Australian coastal waters ($n = 3$).

## DISCUSSION

Increasing evidence suggests that temperate coastal waters may be overlooked hotspots of $N_2$ fixation activity (*Mulholland et al., 2012*; *Mulholland et al., 2019*; *Tang et al., 2019*). Determining the distribution and activity of diazotrophs, and the environmental processes that influence them within coastal zones is therefore important to further our understanding of N availability across diverse marine environments. Compared to previous studies in temperate and tropical estuarine environments, where maximum $N_2$ fixation rates of 30–85 nmol $L^{-1}$ $d^{-1}$ have been observed (*Ahmed et al., 2019*; *Bentzon-Tilia et al., 2015*; *Bhavya et al., 2016*), here we report relatively high rates of $N_2$ fixation in temperate coastal waters of southern Australia within the inverse estuary Spencer Gulf. We show that $N_2$ fixation rates, diazotroph diversity, and community structure, can vary considerably across relatively small spatial scales, however the dynamics of $N_2$ fixation were relatively stable across two contrasting seasons. Our findings suggest that $N_2$ fixation, possibly mediated by UCYN-A and non-cyanobacterial diazotrophs, may provide an important source of fixed N to support primary production within the oligotrophic, temperate coastal waters of southern Australia.

### $N_2$ fixation in temperate coastal environments

Recent efforts to determine the importance of $N_2$ fixation as a source of new N within temperate coastal waters have revealed $N_2$ fixation activity in these regions is similar to, and at times higher than, rates reported for tropical and subtropical open ocean environments (*Mulholland et al., 2019*; *Tang et al., 2019*). For example, maximum $N_2$ fixation rates of

65, 130, and 100 nmol $L^{-1}$ $d^{-1}$ have recently been observed in coastal waters of the north-eastern, mid-, and western Atlantic Ocean respectively (*Fonseca-Batista et al., 2019*; *Mulholland et al., 2019*; *Tang et al., 2019*). In environments representing the traditional niche of $N_2$ fixation, such as the North Pacific Subtropical Gyre (NPSG) and the Eastern South Pacific (ESP), maximum $N_2$ fixation rates have been reported to be considerably lower at $\leq 20$ nmol $L^{-1}$ $d^{-1}$ (*Böttjer et al., 2017*; *Gradoville et al., 2017*; *Shiozaki et al., 2017*).

In the temperate coastal waters of southern Australia, we observed relatively high rates of $N_2$ fixation, with a maximum $N_2$ fixation rate of 64 nmol $L^{-1}$ $d^{-1}$. This observation is similar in magnitude to the high $N_2$ fixation rates reported for the tropical oligotrophic seas of northern Australia (*Bonnet et al., 2015*; *Messer et al., 2016*), and is almost double maximum $N_2$ fixation rates previously reported for tropical estuarine systems (31–34 nmol $L^{-1}$ $d^{-1}$; *Ahmed et al., 2019*; *Bhavya et al., 2016*). The lowest rates of $N_2$ fixation (2 nmol $L^{-1}$ $d^{-1}$) occurred in the continental shelf waters. This finding is comparable to observations from other continental shelf ecosystems where $N_2$ fixation rates are typically lower that those observed in sites closer to the coast (*Mulholland et al., 2012*; *Shiozaki et al., 2015a*; *Singh, Gandhi & Ramesh, 2019*). Intermediate rates of $N_2$ fixation (10–45 nmol $L^{-1}$ $d^{-1}$) were measured within Spencer Gulf, and these rates are placed within the upper end of those previously reported for other temperate coastal, and tropical estuarine waters (*Ahmed et al., 2019*; *Bentzon-Tilia et al., 2015*; *Bhavya et al., 2016*; *Mulholland et al., 2012*; *Rees, Gilbert & Kelly-Gerreyn, 2009*; *Shiozaki et al., 2015a*). Importantly, our findings demonstrate that $N_2$ fixation in the temperate waters of southern Australia are similar to, and can exceed, those observed in the NPSG and ESP (*Gradoville et al., 2017*).

While $N_2$ fixation rates demonstrated clear spatial patterns in their magnitude between southern shelf and Spencer Gulf waters, we observed relatively consistent $N_2$ fixation rates across opposing seasons. This is in contrast to previous seasonal observations of $N_2$ fixation from distinct marine environments, where $N_2$ fixation rates are typically higher during spring/summer than autumn/winter and are accompanied by shifts in the abundance of different diazotrophic taxa (*Bentzon-Tilia et al., 2015*; *Böttjer et al., 2017*; *Fernandez et al., 2015*; *Mulholland et al., 2019*). We hypothesised that seasonal differences in $N_2$ fixation rates would occur within Spencer Gulf and shelf waters due to the known seasonality in physico-chemical characteristics, such as temperature, salinity, and dissolved nutrients, which ultimately influence the distribution and activity of marine diazotrophic microorganisms (*Monteiro, Dutkiewicz & Follows, 2011*; *Moore et al., 2013*; *Ward et al., 2013*). However, while limited in replication, we observed relatively stable site-specific physico-chemical conditions between the two contrasting seasons, and no significant differences in the composition of the underlying diazotrophic community. While limited in scope to two time-points, our observations suggest that relatively high $N_2$ fixation rates can be maintained within Spencer Gulf while favourable conditions prevail. In future, increased sampling resolution is required to define the seasonal dynamics of $N_2$ fixation within the temperate coastal waters of southern Australia.

## Regional significance of biological N$_2$ fixation

Our previous research indicated that the pelagic microbial community of Spencer Gulf includes a diverse array of diazotrophic clades (*Messer et al., 2015*). However, the presence of diazotrophic groups cannot solely be used as evidence for the importance of pelagic N$_2$ fixation, as the physiological process is tightly regulated (*Paerl, Crocker & Prufert, 1987*). To the best of our knowledge, our observations of N$_2$ fixation within the pelagic realm of Spencer Gulf represent the first N$_2$ fixation measurements from a temperate inverse estuary. Our N$_2$ fixation rate measurements support our hypothesis that pelagic N$_2$ fixation may provide a supply of fixed N within Spencer Gulf and the southern shelf waters, at considerably high rates relative to tropical and subtropical open ocean environments. In an earlier study, *Middleton et al. (2013)* estimated the influx of bioavailable N (in the form of NO$_3$ and NH$_4$) within Spencer Gulf to be 16.9 kilotonnes yr$^{-1}$, including anthropogenic N sources and mixing of upwelled nutrients from continental shelf waters. This estimate did not include biological N$_2$ fixation as a source of N, using their estimate of the volume of Spencer Gulf (4.58 $\times 10^{14}$ L), the N$_2$ fixation rates measured herein could theoretically contribute an additional 23–149 kilotonnes N yr$^{-1}$, albeit assuming consistent daily N$_2$ fixation rates for a given site. Indeed, an accurate N budget would require extensive additional N$_2$ fixation rates, with the appropriate modifications to the bubble method used to measure N$_2$ fixation (*White et al., 2020*). Nevertheless, based on our estimates, we propose that the process of biological N$_2$ fixation could be one mechanism by which productivity is maintained throughout the region.

It must be noted that the N$_2$ fixation rates presented herein have been corrected to allow for the incomplete dissolution of the $^{15}$N$_2$ gas bubble at 75% of the theoretical for a 24 h incubation (*Großkopf et al., 2012*; *Mohr et al., 2010*). However, recent methodological comparisons suggest no "global factor" exists for rate corrections to the bubble method (*Wannicke et al., 2018*; *White et al., 2020*). Despite the known caveats of the bubble method, this approach was used in the present study due to the predicted highly dissimilar environmental conditions at each site, which would be very difficult to replicate with pre-prepared $^{15}$N$_2$ saturated artificial seawater (*Wilson et al., 2012*). In particular, the observed gradient in ambient temperature and salinity, which determines gas solubility and is accounted for in the rate calculations based on our observations at each site, would be difficult to anticipate ahead of sample collection. We also note that contamination of Sigma-Aldrich commercial $^{15}$N$_2$ gas stocks was reported after our initial study (*Dabundo et al., 2014*). Although we cannot explicitly rule out contamination in the batch of $^{15}$N$_2$ that we used, assuming the mean values for Sigma Aldrich lot SZ1670V reported in Table 1 of *Dabundo et al. (2014)* are consistent across batches, we estimate that potential contamination from $^{15}$NO$_3$, $^{15}$NH$_4$, and $^{15}$N$_2$O, would represent an extremely small proportion of additional $^{15}$N in our incubations, equivalent to a total of 3.2 x 10$^{-7}$ moles. In our experiments, the relative concentration of $^{15}$N gas added was 2.7 $\times$ 10$^{-4}$ moles. Including this estimate of additional $^{15}$N in our trace additions, any potential contamination would inflate our N$_2$ fixation rates by between 0.001–0.079 nmol L$^{-1}$ d$^{-1}$, which is within the lower end of the inferred N$_2$ fixation rates resulting from $^{15}$NH$_4$ contamination for 4.5 L incubations, presented in Table 2 of *Dabundo et al. (2014)*). Moreover, this estimate
is within our calculated standard error of mean $N_2$ fixation rates across triplicate samples (equivalent to 0.05 - 10.77 $nmol^{-1}$ $d^{-1}$). Therefore, any potential contamination would have a negligible effect on the ultimate $N_2$ fixation rates reported herein. In future work, the modified bubble method should be employed, including additional determination of $^{15}N_2$ atom% enrichment of individual incubation bottles and $^{15}N_2$ gas purity, as recently suggested by the scientific community (*Jayakumar et al., 2017*; *Klawonn et al., 2015*; *White et al., 2020*).

### Identifying the key players in coastal $N_2$ fixation

Understanding the abundance and composition of the diazotrophic community underlying $N_2$ fixation activity is important for deciphering the potential impact of newly fixed N to a given region (*Mulholland, 2007*; *Zehr & Kudela, 2011*). For instance, throughout tropical and subtropical open ocean environments, $N_2$ fixation by autotrophic diazotrophs such as *Trichodesmium* sp. will contribute directly to local primary production and may also release recently fixed $N_2$ into the water column to support the growth of non-diazotrophic organisms (*Berthelot et al., 2015*; *Caffin et al., 2018*; *Garcia, Raimbault & Sandroni, 2007*; *Glibert & Bronk, 1994*; *Mulholland, Bronk & Capone, 2004*). On the other hand, symbiotic diazotrophs such as the heterocystous cyanobacterium *Richelia*, which is typically associated with "tropical" phytoplankton species, transfer fixed $N_2$ directly to their eukaryotic phytoplankton host (*Foster et al., 2011*), and therefore contribute to new production and carbon sequestration in regions where they are abundant, such as the NPSG (*Karl et al., 2012*). While the contribution of newly fixed N by non-cyanobacterial diazotrophs is not yet clear (*Turk-Kubo et al., 2014*), their combined high abundances and widespread transcriptional activity in areas of high $N_2$ fixation rates (*Bird & Wyman, 2013*; *Chen et al., 2018*; *Langlois et al., 2015*; *Moisander et al., 2014*), indicate that they could make an important contribution to support primary production in both open ocean and coastal environments.

The diversity of diazotrophic organisms detected in the present study indicates that $N_2$ fixation activity may directly and indirectly support primary production within Spencer Gulf. Within the diverse diazotrophic communities detected, Cluster 1B UCYN-A, and Cluster 1G and Cluster 3 Proteobacteria, dominated diazotroph community profiles. Specifically, we observed high relative abundances of sequences closely related ($\geq 96\%$ AAI) to the symbiotic UCYN-A, in addition to the presumed free-living *Pseudomonas stutzeri* and *Desulfovibrio aespoeensis*, as well as lower relative abundances of the large filamentous tropical cyanobacterium *Trichodesmium erythraeum*. To date, UCYN-A and gammaproteobacterial diazotrophs (related to *Pseudomonas stutzeri*), have consistently been observed within temperate coastal diazotroph communities (*Bentzon-Tilia et al., 2015*; *Mulholland et al., 2019*; *Needoba et al., 2007*; *Shiozaki et al., 2015b*), but they are also key components of subtropical and tropical assemblages (*Bonnet et al., 2015*; *Langlois et al., 2015*; *Moisander et al., 2014*). Our observations provide further support for the global significance of these groups, although it must be noted that the *Pseudomonas stutzeri* OTUs did not cluster with known sequences from the globally distributed Gamma A clade.

While the presence of *Trichodesmium erythraeum* was somewhat unexpected due to its tropical and subtropical distribution (*Capone et al., 2005*), sequences related to *Trichodesmium* sp. have previously been observed at temperate latitudes of the Atlantic and Pacific Oceans (*Mulholland et al., 2019*; *Rivero-Calle et al., 2016*; *Shiozaki et al., 2015a*), and their presence has also been reported in south Australian waters based on microscopic observations (*Paxinos, 2007*). We did not determine the specific activity of *Trichodesmium sp.* within our samples, however, it comprised up to 17% of the diazotroph community at the mid-Gulf site during Austral summer. Owing to its presence and potential importance for both local primary production and N availability, further investigation into the significance of *Trichodesmium sp.* within temperate coastal waters is required.

Consistent with our previous observations of UCYN-A diversity and distribution within Spencer Gulf (*Messer et al., 2015*), we observed differences in the abundances of the open-ocean UCYN-A1 and the coastal UCYN-A2 within and between the southern shelf waters. The emerging sub-lineage UCYN-A4 (*Farnelid et al., 2016*) was also detected in our amplicon sequencing profiles during Austral summer. Due to the similarity between this OTU and the UCYN-A2 qPCR assay of *Thompson et al. (2014)*, we cannot rule out that our qPCR derived abundances do not contain a mixture of the UCYN-A2 and UCYN-A4 sub-lineages (*Farnelid et al., 2016*). Since UCYN-A1, and to a lesser extent UCYN-A2 (possibly A2/A3/A4 sub-lineages), have recently been shown to be highly abundant ($\leq 10^6$ *nifH* copies $L^{-1}$) and reasonably active, fixing $N_2$ at rates of 6 nmol $L^{-1}$ $d^{-1}$ in the cold surface waters of the Western Arctic Ocean (*Harding et al., 2018*), UCYN-A are highly likely to be important mediators of $N_2$ fixation within Spencer Gulf and more broadly across temperate and coastal marine environments.

### What environmental factors influence $N_2$ fixation in temperate southern Australian waters?

Across the global ocean, SST and subsurface minimum dissolved oxygen concentrations have been identified as the major environmental variables influencing pelagic $N_2$ fixation rates (*Luo et al., 2014*; *Tang, Li & Cassar, 2019*). In addition, the availability of dissolved iron, phosphorus, other N sources (*Landolfi et al., 2015*; *Ward et al., 2013*), and grazing by zooplankton (*Wang et al., 2019*), have all been identified as factors shaping the distribution and magnitude of marine $N_2$ fixation. Although limited in scope and replication, in the temperate southern Australian waters examined here, high $N_2$ fixation rates during the Austral autumn were significantly correlated with increased $PO_4$ concentrations, as was the overall abundance of UCYN-A1 (derived by qPCR). This is consistent with patterns observed in other temperate coastal waters, where $N_2$ fixation has previously been shown to be significantly correlated with phosphorus availability (*Tang et al., 2019*). This pattern is also in-line with patterns observed within more oceanic waters, where $PO_4$ availability has been shown to influence *nifH* expression and $N_2$ fixation rates in experimentally manipulated and natural diazotroph assemblages (*Rees, Law & Woodward, 2006*; *Sañudo Wilhelmy et al., 2001*; *Turk-Kubo et al., 2012*; *Watkins-Brandt et al., 2011*). As phosphorus is an important constituent of cellular and molecular machinery, there is likely a direct causal relationship between $PO_4$ and $N_2$ fixation, whereby diazotroph abundances

and $N_2$ fixation rates are increased under P-replete conditions, as has previously been observed for the UCYN-A1-haptophyte symbiosis (*Krupke et al., 2015*). Within Spencer Gulf, phytoplankton growth is estimated to be limited by $PO_4$ availability year-round (*Middleton et al., 2013*), indicating that while higher $N_2$ fixation rates may provide a source of bioavailable N to the dissolved pool, the increased diazotrophic activity may deplete $PO_4$ concentrations for non-diazotrophic microorganisms.

In the present study, overall $N_2$ fixation rates were also negatively correlated with concentrations of $NO_3/NO_2$, which are typically depleted in Gulf waters during Austral summer yet may remain relatively high on the continental shelf due to a permanent deep nutrient pool (*Doubell et al., 2018*). Spencer Gulf and the adjacent continental shelf waters are characterised by a unique combination of oceanographic and regional circulation processes that create seasonal and localised east–west gradients in ambient concentrations of key macro- and micro-nutrients, underpinning variability in microbial productivity (*Doubell et al., 2018*; *Middleton et al., 2013*; *Van Ruth et al., 2018*). During Austral autumn, the density front at the entrance to Spencer Gulf begins to break down and an influx of continental shelf water, relatively rich in macronutrients, enters the Gulf along the western edge, while the oligotrophic Gulf water exits from the eastern side of the mouth (*Middleton & Bye, 2007*). These north-south and east–west gradients in $NO_3/NO_2$ and $PO_4$ concentrations (low to relatively high, respectively) (*Middleton et al., 2013*), may explain the observed correlations between $N_2$ fixation rates and these nutrients. This suggests that increased $N_2$ fixation activity may occur due to the low concentrations of bioavailable N, further indicating that N derived from $N_2$ fixation could sustain productivity within the N limited Spencer Gulf region. Recently, $N_2$ fixation by UCYN-A was shown to occur even when dissolved inorganic nitrogen sources are replete, and may even be stimulated by increased $NO_3$ concentrations (*Mills et al., 2020*), highlighting the complexity of factors governing $N_2$ fixation activity in the environment. Collectively, our observations in fact represent the classic nutrient regime within which diazotrophs gain a competitive advantage over non-diazotrophic microorganisms (*Ward et al., 2013*), utilising excess $PO_4$ and fixing $N_2$ to support growth.

While $PO_4$ and $NO_3/NO_2$ were correlated with rates of $N_2$ fixation at the sites examined in this study, they were not significant predictors of diazotroph assemblage structure. Rather, the structure of the underlying diazotroph community was significantly influenced by the prevailing salinity and $SiO_4$ concentrations. Regional variability in $SiO_4$ concentrations may reflect abiotic indicators of different water masses, and may drive distinct differences in the composition of microbial assemblages (*Foster et al., 2007*; *Weber et al., 2017*). The observed transition towards increased non-cyanobacterial diazotrophs in the upper shallow waters of the Gulf could be indicative of their redistribution from the sediment or seagrass microbiome (*Brown, Friez & Lovell, 2003*; *Lehnen et al., 2016*), and warrants further exploration of their specific activity, source and contribution to N cycling in Spencer Gulf.

Salinity is a major structuring factor for estuarine microbial communities, driving the transition from freshwater- to marine-adapted lineages (*Bouvier & Del Giorgio, 2002*; *Jeffries et al., 2016*; *Kirchman et al., 2005*), and influencing rates of biogeochemical nutrient

cycling (*Bernhard et al., 2007*; *Bhavya et al., 2016*). Unlike classical estuaries, inverse estuaries such as Spencer Gulf experience hypersaline conditions at the head of the estuary and marine salinities at the mouth, which has previously been shown to influence the overall composition of specific cyanobacterial ecotypes (*Messer et al., 2015*). In the present study, hypersaline regions of Spencer Gulf were associated with an increase in the relative abundance of non-cyanobacterial diazotrophs and a decrease in the abundance of UCYN-A at sites with salinities $>\sim 37$ PSU, which may reflect an inhibitory effect of high salinity on UCYN-A and it's eukaryotic host. In contrast, members of the deltaproteobacteria, related to the Cluster 3 diazotrophs observed in the present study, have previously been shown to be moderately halophilic (*Gam et al., 2009*; *Warthmann et al., 2005*), and their increased relative abundances at the northern most stations of Spencer Gulf suggests they are likely to be halotolerant.

## CONCLUSIONS

This study provides further evidence that marine $N_2$ fixation is not limited to tropical and subtropical open ocean environments, yet is widespread throughout diverse, temperate ecosystems, which have previously been overlooked as hotspots of $N_2$ fixation activity. Our results indicate that $N_2$ fixation is influenced by an interplay of physical and chemical environmental variables, which may have direct and indirect effects on the distribution and activity of diazotrophs in coastal waters. Our data revealed notable stability in $N_2$ fixation across contrasting seasons, suggesting that the oligotrophic conditions of southern Australian coastal waters promote diazotrophy within the region. Notably, our findings suggest that pelagic $N_2$ fixation, mediated by UCYN and non-cyanobacterial diazotrophs, could provide a greater source of fixed N than upwelled and anthropogenic bioavailable N within the coastal waters of southern Australia.

## ACKNOWLEDGEMENTS

Sampling at NRSKAI and SAM8SG was facilitated by the Integrated Marine Observing System (IMOS). We acknowledge the captain and crew of the South Australian Research and Development Institute (SARDI) *RV Ngerin*, in particular Paul Malthouse, for their expertise and assistance with sampling during this study. Data was sourced from IMOS, a national collaborative research infrastructure, supported by the Australian Government.

### Funding

This research was supported by the Australian Research Council Discovery Project scheme, grant number DP120102764, awarded to Mark V. Brown and Justin R. Seymour. The funders had no role in study design, data collection and analysis, decision to publish, or preparation of the manuscript.

### Grant Disclosures

The following grant information was disclosed by the authors:
Australian Research Council Discovery Project scheme: DP120102764.

### Competing Interests

The authors declare there are no competing interests.

### Author Contributions

- Lauren F. Messer conceived and designed the experiments, performed the experiments, analyzed the data, prepared figures and/or tables, authored or reviewed drafts of the paper, and approved the final draft.
- Mark V. Brown, Mark Doubell and Justin R. Seymour conceived and designed the experiments, performed the experiments, authored or reviewed drafts of the paper, and approved the final draft.
- Paul D. Van Ruth conceived and designed the experiments, authored or reviewed drafts of the paper, and approved the final draft.

### DNA Deposition

The following information was supplied regarding the deposition of DNA sequences:

The sequences are available at the NCBI Sequence Read Archive: PRJNA294637, SAMN14858932-SAMN14858941.

NifH sequences data are available at FigShare:

- Messer, Lauren (2020): NifH qPCR/RTqPCR and environmental data from a temperate inverse estuary. figshare. Dataset. https://doi.org/10.6084/m9.figshare.12344312.v1.

### Data Availability

Raw data are available in the Supplemental Files and at FigShare:

- Messer, Lauren (2020): NifH sequences from a temperate inverse estuary in southern Australia. figshare. Dataset. https://doi.org/10.6084/m9.figshare.12279743.v1

- Messer, Lauren (2020): NifH qPCR/RTqPCR and environmental data from a temperate inverse estuary. figshare. Dataset. https://doi.org/10.6084/m9.figshare.12344312.v1

- Messer, Lauren (2020): N2 fixation raw and rate data from Spencer Gulf South Australia. figshare. Dataset. https://doi.org/10.6084/m9.figshare.12961097.v1

### Supplemental Information

Supplemental information for this article can be found online at http://dx.doi.org/10.7717/peerj.10809#supplemental-information.

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
