# Peer review of "Temperate southern Australian coastal waters are characterised by surprisingly high rates of nitrogen fixation and diversity of diazotrophs"

_PeerJ, doi:10.7717/peerj.10809_

## Round 0.1 · original submission · Major Revisions

Two experts have reviewed your manuscript and both agree that it presents novel findings regarding diazotroph activity and composition in a unique estuary setting that will be of interest to the research community. Important questions/suggestions were raised by both. In particular, there are questions about experimental and analytical approaches (e.g., N2 fixation assay, RT-qPCR, sequence analysis of UCYN-A sub-lineages, etc.). Additionally, there were questions concerning the length of the discussion, particularly with regard to the correlation between water chemistry and diazotroph distribution. Reviewer #1 provides a number of other excellent suggestions, including presenting data shown in Figure 1B,C in table format.

Also consider these points:

1. NOx as an abbreviation for nitrate and nitrate is confusing - as it is typically used to designate nitrogen oxide gasses such as NO and NO2. Please change abbreviation or simply use NO2-/NO3- throughout.
2. Detection of significant proportions of nifh belonging to putative anaerobes (i.e. Desulfovibrio, etc.) raise interesting questions regarding their origin and regions of activity. Since freshwater inputs are so low in this estuary, do they arrive from dust storms (is there any literature regarding nutrient or microbe inputs into the estuary via this route)? Or, is it possible that they inhabit sediments in the seagrass regions as shown for Spartina? If they inhabit seagrasss rhizospheres, it has important implication for the estuary’s nutrient dynamics. Once transported to the water column, these anaerobes would likely be inactive, unless attached to relatively large particles where anaerobic microniches are possible. Discussion is warranted.
3. Information about Spencer Gulf is important. Some of the information presented in lines 555-575 could be presented in the introduction.
4. Using the ratio between quantitative reverse transcriptase-PCR and qPCR values to determine transcript numbers can be suspect. Between its reliability, the questions posed by reviewer #1 and the overall impact of these results, you may wish to remove that data and accompanying discussion.

Most of the comment are aimed at improving the manuscript rather than questioning the major findings or quality. Overall a very nice study. I look forward to reading a revised manuscript, if you choose to resubmit.

Reviewer 1 ·

Basic reporting

The writing is clear. The introduction is setting the stage for the study sufficiently.
Literature cited is extensive although a few key ones seem to be missing (see below).
Figure 1 seems low resolution. Some revision would improve it (see below).
The N2 fixation raw and calculated data should be included.
The sequences need to be submitted to the appropriate sequence database. qPCR assay limits of detection and quantification should be reported.
The manuscript is self-contained.

Experimental design

There is a continuous interest in understanding distributions of diazotrophs in the world’s oceans. If the authors can resolve the methodological issues detailed below, this study is adding to the global database of N2 fixation rates and parallel diazotroph communities, providing data from yet another area from which these measurements have not yet been made. The research is essentially a monitoring study. The time of day should be reported for sample collection. If the time varied substantially across samples, the expression data are not going to be comparable (because UCYN-A nifH expression has a diel cycle).
Technical methods for sequencing and qPCR seem ok, although a few additional details should be added.
For the N2 fixation methods, a few other details need to be provided. Importantly, the batch of 15N2 used is the same one used in a previous study demonstrating 15N-DIN contamination in the gas. The authors need to be able to show gas contamination is not influencing their results.

Validity of the findings

The data reported here primarily have value when exploring the importance of coastal areas in general as contributors to N2 fixation and global distributions of UCYNA lineages. Explanations of these patterns with a few environmental snapshot measurements is correlational. n=1 for each environmental datapoint. Multiparameter analyses could further elucidate the patterns explained in the text.

The discussion feels too long and speculative given the scope of study. I wish the focus was on providing more new information and less of a literature review.

Additional comments

L89 The diazotrophs that are not cyanobacteria are not necessarily only heterotrophic. By assuming that all nifH sequences that are not falling with cyanobacteria are from heterotrophic bacteria, you extrapolate from the nifH genes to metabolisms which are unknown. This is a common fallacy in the diazotroph literature.

L104 Need references for the fact that UCYN-B and C are free-living.

L130 You have mentioned UCYN-A several times by now. Define with the full name when mentioning it for the first time.

L131 Include appropriate references for Gamma A. Did you find it here? Should come back to it in the discussion if you did.

L173-174 Were the dissolved nutrient samples filtered for particulates only after freezing, not filtered prior to freezing? If so, can you show this approach did not bias your data?

L197 The same lot of Sigma 15N2 gas has previously been shown to contain 15N-DIN contaminants (Dabundo et al. 2014). This is a major concern. Did you make any attempts to check the gas contamination?

L198 Were any measurements made on actual gas enrichment? What was the final calculated gas enrichment?

L204 At what time of the day did you conduct the incubations? How much did the initiation and termination times vary across stations and sampling times? If this was highly variable, it could also bias comparisons among sites/seasons.

L300-301 Why were PCR conditions different for the A1 and A1?

L357 But did your gas contain 15N-DIN contaminants?

L348-439 I don’t agree with these statements. Salinity and silica could equally well be indicators of the different water masses in which multiple other factors may have been different, correlating with salinity and silica, and played greater roles. You have no way of knowing these two that you happened to measure were the drivers for the changing communities.

L449 Amplicon based approaches should not be interpreted quantitatively. They are always relative abundances. Given the fact that the rest of the community may be changing drastically, same proportion of an OTU in the community based on amplicon sequencing analyses may mean the actual abundances increased, decreased, or remained stable. Therefore comparisons between amplicon sequencing and qPCR should be worded differently.

L458-459 You don’t discuss the time of day for measurements of nifH expression. At what time did you collect the RNA samples? Was the time of day identical across samples? UCYN-A nifH expression has a diel pattern, therefore the results are only comparable across sites and seasons if the collection time was the same.

L474 Why is change in salinity at these ranges potentially driving the communities?

L 481 Also see Mulholland et al. 2012.

L486-488 These rates should only be reported if you can demonstrate your batch of 15N was not contaminated with 15N-DIN. The raw 15N2 data should be included, with data from blanks shown. Did you include any controls incubated without tracer to check if background shifted during incubation? Was the background delta 15N measured for T0 for each sample? Data for any of such controls should be reported.

L546 Physico-chemical data being based on n=1 per site per sampling occasion and fairly limited.

L675-768 This part is essentially literature review, and very long speculation with your small dataset. The literature review seems fine, but overextended. All of your data are correlative - it is difficult for me to get excited about this part.

L712-713 Mills et al. 2020, ISME J provides insights into DIN and UCYNA. Their findings are important for your conclusions (as they appear to contradict them).

L760-763 The statement about influence of salinity on freshwater diazotrophic cyanobacteria here seems like a stretch. You studied completely different species in a marine environment. The link to your study seems weak.

L765 State as a conclusion from your data, not as a hypothesis at this late stage of the narrative.

Figures

Figure 1.The different nutrients should be separated into different graphs or shown in a table, as they require different scales to see any changes.
You measured particulate C and N also, where are the data?

Figure 3. Please separate the UCYNA1 and A2 OTUs.

Figure 4. Did you check assay cross-hybridization for UCYN-A1 and A2 under your assay conditions?

Figure 5. What time of the day were cDNA collected? The data are comparable only if the sampling time was identical. Different colors for different symbols would help here.

·

Basic reporting

The manuscript by Messer et al. report nitrogen fixation rates, diazotroph community composition and quantification of UCYN-A (A1 + A2/A3 sublineages) in an inverse estuary. The findings add on to the literature on the significance of nitrogen fixation also in coastal temperate waters. I think that this is a very nice area to study shifts in diazotroph communities and activity. The manuscript is very well written and the data is clearly presented.

Experimental design

My main criticism concerns the presentation of data related to sublineages of UCYN-A. In the current study the authors have used an OTU clustering approach of 97% similarity. The cluster is then identified by the most abundant sequence type in the cluster which the authors use to identify the cluster. The problem with this method is that UCYN-A sublineages are >97% similar on the amino acid level and the UCYN-A diversity is therefore “hidden” in the clustering. This needs to be revised or discussed more thoroughly, especially since different UCYN-A sublineages are targeted in the qPCR analysis which cannot be identified on the OTU level in the amplicon libraries. To use the full potential of the data, analysis of amplicon sequence variants would have been a better choice.

Further, in literature it is well documented that the UCYN-A2 qPCR assay by Thompson et al. 2014 also amplifies the UCYN-A sublineage UCYN-A3 which frequently co-occurs with UCYN-A1 (see Turk-Kubo et al. 2017 and Farnelid et al 2016 for evaluation of qPCR primers targeting UCYN-A). In the current study, with lack of support from the amplicon libraries it is not possible to conclude whether UCYN-A2 and/or UCYN-A3 is co-occurring with UCYN-A1.

It is also not clear if Gamma A sequences were present in the study? This would be interesting to know since it is one of the most well documented and widely distributed HBN’s.

Validity of the findings

The discussion on correlations between SiO4 concentrations and diazotroph community structure (Lines 731-746) is highly speculative as SiO4 may be linked to abiotic factors rather than the occurrence of diatoms and neither have been explored in this study. The authors draw the conclusion that cyanobacterial diazotrophs associated with diatoms were not important constituents of the diazotroph communities (Lines 737-739). It should be noted that the primers used in this study have been shown to favor amplification of gammaproteobacterial taxa while systematically underrepresenting Richelia and Calothrix (Turk-Kubo et al. 2015). Thus in order to draw conclusions about the occurrence of symbionts of diatoms analyses need to be coupled to microscopy observations or complemented by qPCR analyses specific to these groups.

Additional comments

Farnelid, H., Turk-Kubo, K., Muñoz-Marín, M., and Zehr, J. (2016) New insights into the ecology of the globally significant uncultured nitrogen-fixing symbiont UCYN-A. Aquat Microb Ecol 77: 125–138.
Turk-Kubo, K.A., Farnelid, H.M., Shilova, I.N., Henke, B., and Zehr, J.P. (2017) Distinct ecological niches of marine symbiotic N2-fixing cyanobacterium Candidatus Atelocyanobacterium thalassa sublineages. J Phycol 53: 451–461.
Turk-Kubo, K.A., Frank, I.E., Hogan, M.E., Desnues, A., Bonnet, S., and Zehr, J.P. (2015) Diazotroph community succession during the VAHINE mesocosm experiment (New Caledonia lagoon). Biogeosciences 12: 7435–7452.

---

## Round 0.2 · Minor Revisions

Two reviewers have read your re-submission and both find the manuscript improved, and think it will be a nice addition to our understanding of diazotroph distribution and activity in estuaries. They both have several minor suggestions that should be addressed before publication. In particular, addressing reviewer #1's comments regarding the 15N contaminant issue through additional text should help to alleviate readers concerns. This reviewer #1 is different from the original reviewer #1, thus the issue of 15N contamination is obviously a touchy issue. I leave it up to the authors to decide if their "discussion on SiO4 concentration in relation to increased abundances of non-cyanobacterial diazotrophs" is too speculative.

Apologies for the slow turnaround. This editor was overwhelmed over the past month. If you choose to submit a revised manuscript, it will not be necessary to send it out for re-review.

·

Basic reporting

Thank you for the revised manuscript. This study is interesting and an important contribution to the growing litterature on the activity and distribution of unicellular and non-cyanobacterial diazotrophs. The manuscript is generally well written.

In the revised manuscript, multiple punctuation errors such as missing brackets, double full stops, and spelling mistakes (e.g Proteobateria) are present which need to be corrected prior to publication

Experimental design

no comments

Validity of the findings

Similar to my comment in the previous review, I still find the section in the discussion on SiO4 concentration in relation to increased abundances of non-cyanobacterial diazotrophs highly speculative. A concentration of 1.1 uM SiO4, as observed in the waters of the Gulf should not be described as "rich" as similar concentrations represent very low post diatom bloom levels of SiO4 in oligotrophic waters (e.g. Sargasso Sea). Considering the overall low concentrations of SiO4 in the Gulf, I encourage a revision or removal of this section.

·

Basic reporting

This manuscript presents interesting findings of N2 fixation rates and diazotrophic community structure in an inverse estuary in Southern Australia. The authors’ findings of relatively high N2 fixation rates adds to the mounting evidence that N2 fixation may be more important in coastal environments than was previously recognized. The manuscript is very well-written and clear.

Experimental design

My only concern about the experimental design is the potential for 15N contaminants in the Sigma batch of 15N2 gas used, which could have inflated N2 fixation rate estimates (as discussed by Reviewer #1 in the previous round of review). The authors now discuss this issue in lines 589-602 of the current manuscript. I feel that a slightly longer description of the calculations/assumptions used to estimate the potential inflated N2 fixation rates (referencing Table 2 of Dabundo et al. (2014)), would help convince the reader that contamination was not an issue.

Validity of the findings

The manuscript conclusions are well-stated and supported by the data.

Additional comments

There are a few instances of small grammatical/formatting errors (L205, 217, 486-488, 523-526)

L486-497: Were UCYN-A abundances significantly correlated with N2 fixation rates?

The authors may wish to discuss the fact that UCYN-A qPCR abundances were relatively low while N2 fixation rates were quite high. I suspect that a quick calculation multiplying maximum UCYN-A cell-specific N2 fixation rates from the literature by the observed qPCR abundances would indicate that UCYN-A likely accounts for a small fraction of the observed N2 fixation rates. However, this method has huge caveats, including that nifH gene abundances are not equivalent to cell concentrations; it is possible that nifH underestimates cells due to incomplete DNA extraction or other methodological issues. Whether or not this should be discussed can be at the discretion of the authors.

---

## Round 0.3 · accepted · Accept

Thank you for addressing the reviewers minor concerns and your patience during the longer than normal review process. Your manuscript will be a welcome addition to the limited research on N fixation in estuary ecosystems.